# Perioperative Probiotics Application for Preventing Postoperative Complications in Patients with Colorectal Cancer: A Systematic Review and Meta-Analysis

**DOI:** 10.3390/medicina58111644

**Published:** 2022-11-14

**Authors:** Sanghyun An, Kwangmin Kim, Myung Ha Kim, Jae Hung Jung, Youngwan Kim

**Affiliations:** 1Department of Surgery, College of Medicine, Yonsei University Wonju, Wonju 26426, Korea; 2Center of Evidence Based Medicine, Institute of Convergence Science, Yonsei University, Seoul 03722, Korea; 3Yonsei Wonju Medical Library, College of Medicine, Yonsei University Wonju, Wonju 26426, Korea; 4Department of Urology, College of Medicine, Yonsei University Wonju, Wonju 26426, Korea

**Keywords:** colorectal neoplasm, probiotics, synbiotics, postoperative complications

## Abstract

*Background and Objectives*: Perioperative probiotic administration in patients who undergo gastrointestinal surgery can reduce postoperative infectious complications. This systematic review and meta-analysis aimed to evaluate the effect of probiotics on postoperative outcomes in patients who underwent colorectal cancer surgery. *Materials and Methods*: For this study, we followed the protocol published by PROSPERO (registration number: CRD42021247277). We included studies on patients undergoing open, laparoscopic, or robotic colorectal cancer surgery for curative intent. We conducted a comprehensive search with online databases (trial registries and ClinicalTrials.gov), other literature sources, and conference proceedings, with no language restriction, up until 12 August 2022. We assessed risk of bias, extracted data, and conducted statistical analyses by using a random-effects model and interpreted the results based on the Cochrane Handbook for Systematic Reviews of Interventions. We rated the certainty of evidence (CoR) according to the GRADE approach. *Results*: We identified 20 published full-text studies. The use of probiotics probably results in little to no difference in perioperative mortality (risk ratio (RR): 0.17, 95% CI: 0.02 to 1.38; I^2^ = 0%; moderate CoE) and may result in reducing the overall postoperative infectious complications (RR: 0.45, 95% CI: 0.27 to 0.76; I^2^ = 38%; low CoE) after colorectal cancer surgery. Probiotics may result in little to no difference in probiotics-related adverse events (RR: 0.73, 95% CI: 0.45 to 1.19; I^2^ = 0%; low CoE). While probiotics may result in reducing the overall postoperative complications (RR: 0.47, 95% CI: 0.30 to 0.74; I^2^ = 8%; low CoE), it may result in little to no difference in hospital length of stay (LOS) (MD: −1.06, 95% CI: −1.64 to −0.47; I^2^ = 8%; low CoE) and postoperative quality of life (QOL) (MD: +5.64, 95% CI: 0.98 to 10.3; low CoE). *Conclusions*: Perioperative probiotic administration may reduce complications, including overall infectious complications, in patients undergoing colorectal cancer surgery without any additional adverse effects. In addition, probiotics may have similar effects on perioperative mortality; procedure-related complications such as anastomotic leakage, and hospital LOS; or improve the QOL. Thus, probiotics may be considered a beneficial supplement to routine perioperative care for colorectal cancer surgery.

## 1. Introduction

Worldwide, colorectal cancer (CRC) is the third most common malignancy and the second leading cause of cancer-related mortality [1]. There has been improvement in the outcomes of various treatment modalities, including surgery, chemotherapy, radiation therapy, targeted therapy, and immunotherapy, for patients with colorectal cancer. Specifically, radical resection is crucial for treatment; additionally, the completeness of oncologic resection is a crucial factor affecting the prognosis [2,3]. However, colorectal surgery has higher postoperative infectious complications than other surgery types given the direct surgical manipulation and preoperative preparation strategies of the large intestine, where bacteria are most abundant [4,5]. Postoperative complications not only directly worsen the patient’s condition but also delay or impede further treatment, such as adjuvant chemotherapy, which can adversely affect the oncological long-term prognosis. Although prophylactic antibiotics are administered to reduce postoperative infectious complications, the rate of infectious complications remains high after colorectal surgery; additionally, inappropriate antibiotic use may cause several adverse effects [6].

There is increasing interest in the influence of gut microbiota on human immunity [4,7]. Gastrointestinal surgery, including colorectal surgery, alters the gut microbiota due to surgical trauma; additionally, microbiota changes and intestinal barrier damage may cause systemic inflammation and promote the development of various chronic diseases, including cancer [8,9,10]. Probiotics, which are defined as microorganisms, can be used to modulate gut microbiota and exert beneficial effects on the host [11]. Perioperative probiotic administration in patients undergoing gastrointestinal surgery has been shown to reduce postoperative infectious complications [12,13]. Furthermore, several trials have shown that perioperative probiotic administration during colorectal surgery effectively reduces infectious complications [14,15,16,17,18]. Additionally, probiotic administration can reduce the levels of inflammatory markers and cytokines [14,19]. However, there have been inconsistent reports regarding the effects of probiotics on patients undergoing colorectal cancer surgery, which could be attributed to differences in the study design and probiotic usage across randomized controlled trials (RCTs). Although several systematic reviews have been conducted on the effectiveness of probiotics, the certainty of evidence (CoE) of the outcomes remains unclear and some of the systematic reviews included heterogeneous participants.

Therefore, we aimed to conduct a systematic review and meta-analysis by using more thorough inclusion criteria for RCTs and including more recent and reliable findings to evaluate the effect of probiotics on postoperative outcomes in patients who underwent colorectal cancer surgery. Additionally, we aimed to evaluate the level of evidence for major outcomes, using the GRADE methodology (Grading of Recommendations, Assessment, Development, and Evaluation).

## 2. Materials and Methods

### 2.1. Literature Search

This study was performed by following the protocol published by PROSPERO (registration number: CRD42021247277). Furthermore, this systematic review followed the Preferred Reporting Items for Systematic Reviews and Meta-analyses (PRISMA) guidelines (Appendix A).

We performed a comprehensive search of several databases, including MEDLINE; EMBASE; Cochrane Library; Scopus; Web of Science; Latin American and Caribbean Health Sciences Literature; and other resources, including ClinicalTrials.gov (www.clinicaltrials.gov/, accessed on 1 November 2022), the World Health Organization International Clinical Trials Registry Platform search portal (apps.who.int/trialsearch/, accessed on 1 November 2022), and OpenGrey (www.opengrey.eu/, accessed on 1 November 2022). The search terms included “colorectal neoplasm”, “probiotics”, “synbiotics”, “lactobacillus”, “Bifidobacterium”, “lactococcus”, “saccharomyces”, “Enterococcus”, “Pediococcus”, “Cultured milk products”, and “streptococcus”. Appendix A illustrates the detailed search strategy for each database. Moreover, we searched the reference lists of the selected studies for supplemental studies, as well as contacted their authors for reports of unpublished or published studies, including new or progressing studies.

The date of the initial search of all the databases was 26 May 2021, and the latest search was performed on 12 August 2022. We identified and removed potentially duplicated records, using reference management software (EndNote, version 20, Clarivate Analytics, Boston, MA, USA). Two authors (S.A. and K.K.) independently screened all relevant records and classified them based on the criteria provided in the Cochrane Handbook for Systematic Reviews of Interventions [20]. Screening was performed by using Rayyan, which is a web and mobile application for systematic reviews (available at www.rayyan.ai, accessed on 1 November 2022). We resolved disagreement by discussion. We included parallel-group RCTs and considered cluster RCTs for inclusion, with no restriction of the publication status or language. We excluded non-applicable crossover studies and nonrandomized studies.

### 2.2. Characteristics of Participants

Eligible participants comprised patients undergoing open, laparoscopic, or robotic colorectal cancer surgery for curative intent. We excluded trials that included patients with unresectable advanced disease, patients who underwent concomitant resection of other organs, patients with co-occurrence of other malignant neoplasm or gastroenterological diseases, patients who underwent emergency surgery, patients who recently received antibiotics therapy for other infectious diseases, and patients with inoperable disease due to comorbidities. Additionally, we only included RCTs performed on patients with colorectal malignancy and excluded RCTs wherein the proportion of other diseases was >10% from the meta-analysis.

### 2.3. Types of Interventions and Comparators

We compared postoperative outcomes between patients with and without perioperative probiotic administration during colorectal cancer surgery. To ensure fair comparisons, concomitant interventions had to be similar between the experimental and comparator groups. The experimental interventions included any type of probiotics, synbiotics, a mixture of probiotics, and prebiotics. The comparators were patients who received placebo or standard care, without any other interventions.

### 2.4. Types of Outcomes

We did not use the measurement of the outcomes assessed in this review as an eligibility criterion.

### 2.5. Primary Outcomes

The primary outcome measures included perioperative mortality, postoperative infectious complications, and probiotics-related adverse events. Perioperative mortality was defined as any death, regardless of cause, occurring within 30 postoperative days. Overall postoperative infectious complications were defined as any infectious complications occurring within 30 postoperative days, and we tried to collect information about the Clavien–Dindo classification. We included clinically confirmed and reported complications in each RCT. Probiotics-related adverse events were defined as unexpected symptoms appearing after taking probiotics, including mild discomfort to discomfort severe enough to stop probiotic administration.

### 2.6. Secondary Outcomes

Secondary outcome measures included overall postoperative complications, hospital length of stay (LOS), and postoperative quality of life (QOL). Overall postoperative complications comprised both infectious and non-infectious complications. Hospital LOS was defined as the period from the day of surgery to the day of discharge. Postoperative QOL was assessed based on the questionnaire used in each study, including the Gastrointestinal Quality of Life Index (GIQLI), European Organization for Research and Treatment of Cancer-Quality of life questionnaire 30, and The Short Form (36) Health survey (SF-36).

### 2.7. Assessment of Risk of Bias

Two review authors (S.A. and K.K.) assessed the risk of bias of each included study independently. We resolved disagreements by consensus or by consultation with a third researcher (J.H.J.). The risk of bias was assessed by using the Cochrane risk of bias tool for randomized trials. The risk-of-bias domains were “low risk”, “high risk”, or “unclear risk”, which were evaluated by using individual items, as described in the Cochrane Handbook for Systematic Reviews of Interventions [21].

### 2.8. Data Collection and Analysis

We extracted outcome data for the calculation of summary statistics and measure of variance. Appendix A shows basic information from the included studies. For dichotomous outcomes, we obtained the number of events and their proportions, as well as the summary statistics with the corresponding measures of variance. For continuous outcomes, we obtained the means, standard deviations, or other necessary data. In the case of continuous outcomes presented as median and range, we sent an email to the corresponding author, requesting the mean value and standard deviation. If we did not receive a response, we converted the values to the mean and standard deviation, using the specified formula [22]. Data were summarized by using a random-effects model and interpreted following the whole distribution of effects. We used the Mantel–Haenszel method and inverse variance method for dichotomous and continuous outcomes, respectively. Statistical analyses were performed by using Review Manager 5 software (The Cochrane Collaboration, Copenhagen, Denmark). The impact of heterogeneity on the meta-analysis was assessed and interpreted based on the guidelines of the Cochrane Handbook for Systematic Reviews of Interventions [20]. There was expected heterogeneity in age (<65 years vs. ≥65 years), tumor location (colon vs. rectal cancer), and neoadjuvant therapy in rectal cancer (with vs. without neoadjuvant chemoradiotherapy); accordingly, we planned to conduct subgroup analyses with an investigation of interactions limited to primary outcomes. Sensitivity analyses of primary and secondary outcomes were only performed for RCTs to explore the influence of placebo on effect sizes, with the exclusion of single-blind studies that only used standard care, without applying a placebo in the control group. However, we could not perform subgroup analyses due to a lack of relevant data and the scarcity of RCTs. If there are more than 10 studies investigating a specific outcome, we used funnel plots to assess small-study effects.

### 2.9. Summary of Findings Table

We presented the overall CoE for each outcome according to the GRADE approach, which takes into account criteria related to internal validity (risk of bias, inconsistency, imprecision, and publication bias) and external validity, such as the directness of results [23]. Two authors (S.A. and K.K.) independently rated the CoE for each outcome, with disagreements being resolved through discussion with a third researcher (J.H.J.).

## 3. Results

### 3.1. Search Results

The database search identified 1851 records; moreover, two additional records were identified from other sources. After removing duplicate records, the titles and abstracts of 1471 records were initially screened, and 1367 records were excluded. Subsequently, we performed full-text screening of 50 articles and excluded 13 studies (16 records) that did not meet the inclusion criteria or were irrelevant to our objectives. Finally, we included 20 RCTs (34 records) in the systematic review. The assessment process is illustrated in the PRISMA flowchart (Figure 1).

### 3.2. Included Studies

Finally, we included 20 published full-text studies [14,15,16,17,18,24,25,26,27,28,29,30,31,32,33,34,35,36,37,38]; among them, 17 and 3 studies were published in English and Chinese, respectively [31,35,36]. We translated papers published in Chinese, using Google Translate. The RCTs were conducted in various countries as follows: China (*n* = 6) [15,31,33,34,35,36], Japan (*n* = 3) [17,28,30], Brazil (*n* = 2) [14,26], Greece (*n* = 2) [18,27], Slovenia (*n* = 2) [29,37], Korea (*n* = 1) [24], Italy (*n* = 2) [32,38], Bosnia (*n* = 1) [25], and Malaysia (*n* = 1) [16]. We attempted to contact all first or corresponding authors and designated a contact person to obtain additional information on the study methods and results, but no one replied. Most of the included RCTs were on patients who had undergone colorectal cancer surgery; furthermore, in one study, 98.89% (358 out of 362) of the participants were patients who underwent surgery for colorectal malignancy [17].

Table 1 shows the baseline characteristics of the included RCTs. Most of the studies were single-center studies, except for three; furthermore, most were performed between 2005 and 2018.

The studies included 1763 randomized participants (*n* = 884, intervention group; *n* = 879, control group). In the intervention group, 524 patients (14 studies) took probiotics, and 325 patients (6 studies) took synbiotics [14,17,26,27,29,37]. In the control group, 453 patients (eight RCTs) received standard care without a placebo [17,25,28,29,30,35,36,37]. All medications, including placebo, were orally administered. The mean ages of the intervention and control groups ranged from 59.8 to 71.5 years and from 58.9 to 72.9 years, respectively. Most of the studies included patients who underwent colorectal cancer surgery, with only four RCTs including patients who underwent colon cancer surgery [24,30,32,37]. Among them, Park et al. only included patients diagnosed with sigmoid colon cancer who underwent anterior resection [24]. Two RCTs only included patients who underwent surgery with laparoscopy [17,32]. The probiotic type and dosage used varied across studies. Seven, three, and ten studies administered probiotics or synbiotics preoperatively, postoperatively, and perioperatively, respectively.

Eight studies reported the perioperative mortality within 30 days after index surgery [14,15,16,17,25,31,34,37]. Overall postoperative infectious complication was reported in seven studies [14,18,24,30,34,35,37]. Seven studies reported the probiotics-related adverse events [14,15,17,24,31,32,35]. Six studies reported the overall postoperative complications [14,16,18,24,32,37]. Eight studies reported the hospital LOS [14,15,16,28,31,32,34,37]. Postoperative QOL was reported in three RCTs [24,27,32]. Theodoropoulos et al. [27] analyzed postoperative QOL by using the GIQLI and EORTC QLQ-C30. They investigated the GIQLI domains of global, symptoms, emotional, physical, and social functions, as well as constipation and diarrhea in EORTC QLQ-C30. Since the GIQLI global score was calculated by summarizing the points of 36 questions, it was included in the meta-analysis. Park et al. investigated the postoperative QOL by using EORTC QLQ-C30 [24]. Pellino et al. surveyed postoperative QOL, using the SF-36 questionnaire weekly for 4 weeks.

### 3.3. Excluded Studies

We excluded 13 studies (16 records) after evaluating the full-text articles; among them, 2 studies (3 records) were not RCTs, including a prospective longitudinal study [39] and a retrospective study using data from an RCT conducted for other purposes [40]. Six studies (eight records) included populations that did not meet our criteria [41,42,43,44,45,46]. We excluded studies that included >10% of patients who underwent surgery for a disease other than colorectal cancer [41,42]. In case of unclear relevant details about participants, an inquiry email was sent to the corresponding author, and, in case of no response, the study was excluded [43]. Two studies (two records) were excluded due to the intervention not meeting the set criteria [19,47]. Appendix A presents further details regarding the characteristics of the excluded studies.

### 3.4. Risk of Bias of Included Studies

Figure 2 presents the risk of bias in the included studies. Four RCTs [25,29,32,34] were judged as unclear risk of bias for random sequence generation. Fifteen of the RCTs studies were judged as unclear risk of bias for allocation concealment [15,16,18,24,25,26,28,30,31,32,33,34,35,36,37]. Eight RCTs [17,25,28,29,30,35,36,37] and one RCT [32] were judged as high and unclear risk of bias, respectively, for blinding of participants and personnel. Nine RCTs were judged as unclear risk of bias for blinding of outcome assessment of subjective outcomes (overall postoperative infectious complications, probiotics-related adverse events, overall postoperative complications, and postoperative QOL) [17,25,29,30,31,32,35,36,37]. All studies were classified as having a low risk of bias for blinding-of-outcome assessment of objective outcomes. Five studies were judged as having a high risk of bias for selective reporting since they did not report outcomes described in the material and methods section or protocol in the full-text article [15,16,17,24,35], while ten studies were judged as unclear risk of bias for selective reporting since we could not identify the study protocol [14,25,28,29,31,32,33,34,36,37]. Two [26,33] and three [25,29,37] studies were judged as high risk of bias for other biases due to differences between the study protocol and the content of the published article and lack of information regarding baseline characteristics, respectively.

### 3.5. Effects of Interventions (Table 2)

#### 3.5.1. Primary Outcomes (Table 2)

##### Perioperative Mortality

Eight studies with 753 participants (intervention: 363; control: 390) were analyzed for perioperative mortality [14,15,16,17,25,31,34,37]. Among them, five mortality events occurred only in the control group in two RCTs [14,16]. Probiotics probably result in little to no difference in perioperative mortality (risk ratio (RR): 0.17; 95% CI: 0.02 to 1.38; I^2^ = 0%; moderate CoE). We downgraded the CoE for serious imprecision.

**Table 2 medicina-58-01644-t002:** Summary of findings.

Probiotics Compared to Placebo or No Treatment; Primary and Secondary Outcomes for Postoperative Outcome					
Patient: Colorectal Cancer Patients Who Underwent Curative Resection Setting: Inpatient Intervention: Probiotics Comparison: Placebo or Standard Care					
Outcomes	No of Participants (Studies) Follow-Up	Certainty of the Evidence (GRADE)	Relative Effect (95% CI)	Anticipated Absolute Effects	
				Risk with Placebo or No Treatment; Primary Outcomes	Risk Difference with Probiotics
Perioperative mortality follow-up: 30 days MCID: 2% absolute difference	753 (8 RCTs)	⨁⨁⨁◯ Moderate ^a,b^	RR 0.17 (0.02 to 1.38)	13 per 1000	11 fewer per 1000 (13 fewer to 5 more)
Overall postoperative infectious complication follow-up: 30 days MCID: 5% absolute difference	651 (7 RCTs)	⨁⨁◯◯ Low ^b,c^	RR 0.45 (0.27 to 0.76)	252 per 1000	138 fewer per 1000 (184 fewer to 60 fewer)
Probiotics related adverse events follow-up: 30 days MCID: 5% absolute difference	692 (7 RCTs)	⨁⨁◯◯ Low ^b,c^	RR 0.73 (0.45 to 1.19)	70 per 1000	19 fewer per 1000 (38 fewer to 13 more)
Overall postoperative complications follow-up: 30 days MCID: 5% absolute difference	394 (6 RCTs)	⨁⨁◯◯ Low ^b,c^	RR 0.47 (0.30 to 0.74)	359 per 1000	190 fewer per 1000 (251 fewer to 93 fewer)
Hospital length of stay follow-up: 30 days MCID: 2 days ^e^	411 (8 RCTs)	⨁⨁◯◯ Low ^b,c^	-	The mean hospital length of stay ranged from 4 to 23 days	MD 1.06 days lower (1.64 lower to 0.47 lower)
Quality of Life Scale from: 0 to 144 follow-up: 1 months MCID: 6.5 points ^f^	67 (1 RCT)	⨁⨁◯◯ Low ^d^	-	The mean quality of Life was 71.36	MD 5.64 higher (0.98 higher to 10.3 higher)

The risk in the intervention group (and its 95% confidence interval) is based on the assumed risk in the comparison group and the relative effect of the intervention (and its 95% CI). CI, confidence interval; MD, mean difference; RR, risk ratio; MCID, minimal clinically important difference; RCT, randomized controlled trial. GRADE Working Group grades of evidence. High certainty: We are very confident that the true effect lies close to that of the estimate of the effect. Moderate certainty: We are moderately confident in the effect estimate: the true effect is likely to be close to the estimate of the effect, but there is a possibility that it is substantially different. Low certainty: Our confidence in the effect estimate is limited; the true effect may be substantially different from the estimate of the effect. Very low certainty: We have very little confidence in the effect estimate; the true effect is likely to be substantially different from the estimate of effect. ^a^ Not downgraded for study limitation: There were two studies in which a mortality event occurred, and no study limitation was observed in the two included studies. Six other studies didn’t have any mortality events. ^b^ Downgraded one level for imprecision: optimal information size was not met. ^c^ Downgraded one level for study limitation: allocation was clearly not concealed in most of the studies, and/or participants were clearly not blinded in the studies. ^d^ Downgraded two levels for imprecision: optimal information size was not met and confidence interval crosses assumed threshold of clinically important difference. ^e^ The value was determined based on thorough discussion by clinical experts. ^f^ MCID from Shi et al. [48].

##### Overall Postoperative Infectious Complication

Seven RCTs with 651 participants (intervention: 329; control: 322) were analyzed for overall postoperative infectious complications [14,18,24,30,34,35,37]. Postoperative infectious complications were observed in 43 and 81 patients in the intervention and control groups, respectively. Probiotics may result in reducing overall postoperative infectious complications after colorectal cancer surgery (RR: 0.45, 95% CI: 0.27 to 0.76; I^2^ = 38%; low CoE). We downgraded the CoE for serious study limitations and serious imprecision.

##### Probiotics-Related Adverse Events

Seven RCTs with 692 participants (intervention: 333; control: 359) were analyzed for probiotics-related adverse events [14,15,17,24,31,32,35]. Eighteen adverse events were reported in the intervention group and 25 events in the control group. Probiotics administration may result in little to no difference in probiotics-related adverse events (RR: 0.73; 95% CI: 0.45 to 1.19; I^2^ = 0%; low CoE). We downgraded the CoE for serious study limitations and imprecision.

#### 3.5.2. Secondary Outcomes (Table 2)

##### Overall Postoperative Complications

Six RCTs with 394 participants (intervention: 199; control: 195) were analyzed for overall postoperative complication [14,16,18,24,32,37]. A total of 32 patients in the intervention group and 70 patients in the control group experienced any postoperative complications. Probiotics may result in reducing overall postoperative complications after colorectal cancer surgery (RR: 0.47; 95% CI: 0.30 to 0.74; I^2^ = 8%; low CoE). We downgraded the CoE for serious study limitations and serious imprecision.

##### Hospital LOS

Eight RCTs with 411 participants (intervention: 207; control: 204) were analyzed for hospital LOS [14,15,16,28,31,32,34,37]. The mean hospital LOS ranged from 3 to 21.4 days in the intervention group and 4 to 23 days in the control group. Probiotics may result in little to no difference in hospital LOS after colorectal cancer surgery (MD: −1.06; 95% CI: −1.64 to −0.47; I^2^ = 8%; low CoE). We downgraded the CoE for serious study limitations and serious imprecision.

##### Quality of Life (QOL)

One RCT with 67 participants (intervention: 34; control: 33) was analyzed for the gastrointestinal-function-related quality of life (GIQLI) [27]. The baseline GIQLI global scores were 74.27 and 70.94 in the intervention and control groups, respectively (*p* = 0.17). The 1-month global score of the GIQLI was 77 ± 9.74 and 71.36 ± 9.71 in the intervention and control groups, respectively (*p* = 0.01). Considering a minimal clinically important difference (MCID) of 6.5 points, probiotics may result in little to no difference in the QOL after colorectal cancer surgery (MD: +5.64; 95% CI: 0.98 to 10.3; low CoE). We downgraded the CoE for very serious imprecision.

### 3.6. Sensitivity Analysis: Double-Blinded Placebo-Controlled Studies Only

After excluding eight single-blind studies [17,25,28,29,30,35,36,37] in which the control group only received standard care without a placebo, we analyzed the results of the remaining twelve studies.

#### 3.6.1. Perioperative Mortality

This sensitivity analysis included five studies [14,15,16,31,34]. The RR was 0.17 (95% CI: 0.02 to 1.38; participants = 273; I^2^ = 0%), which was similar to the results of the main analysis (RR: 0.17; 95% CI: 0.02 to 1.38).

#### 3.6.2. Overall Postoperative Infectious Complications

This sensitivity analysis included four studies [14,18,24,34]. The RR was 0.35 (95% CI: 0.21 to 0.60; participants = 356; I^2^ = 0%), which did not alter the effect seen in the main analysis (RR: 0.45; 95% CI: 0.27 to 0.76).

#### 3.6.3. Probiotics-Related Adverse Events

This sensitivity analysis included five studies [14,15,24,31,32]. The RR was 0.78 (95% CI: 0.46 to 1.33; participants = 270; I^2^ = 0%), which was similar to the results of the main analysis (RR: 0.73; 95% CI: 0.45 to 1.19).

We did not perform a sensitivity analysis for secondary outcomes, including overall postoperative complications, hospital LOS, and postoperative QOL, since the studies eligible for subgroup analysis were the same as those in the main analysis. Overall, the analysis of only the placebo-controlled studies yielded similar results as the main analysis, thus indicating that the meta-analysis results were relatively credible.

## 4. Discussion

Our findings demonstrated that probiotics could effectively reduce infectious complications and overall postoperative complications after colorectal cancer surgery; moreover, there was no additional increase of probiotics-related adverse events. Probiotics administration was not associated with mortality within 30 postoperative days. Additionally, there was no clinically significant influence of probiotic administration on the hospital LOS and postoperative QOL.

Our findings showed that probiotics may significantly reduce the overall postoperative infectious complications; we additionally analyzed in RCT included surgical site infection, anastomosis site leakage, intraabdominal abscess, pneumonia, urinary tract infection (UTI), bloodstream infection, and clostridium difficile infection. In our analysis, perioperative probiotics administration influenced the incidence of pneumonia (RR: 0.39; 95% CI: 0.22 to 0.70), UTI (RR: 0.46, 95% CI: 0.23 to 0.93), surgical site infection (RR: 0.65; 95% CI: 0.49 to 0.86), and bloodstream infection (RR: 0.45; 95% CI: 0.26 to 0.77). However, it did not reduce the incidence of anastomosis site leakage (RR: 0.78; 95% CI: 0.45 to 1.33), intra-abdominal abscess (RR: 0.85; 95% CI: 0.52 to 1.34), and clostridium difficile infection (RR: 0.61; 95% CI 0.26 to 1.43). This indicates that probiotics were effective in outcomes related to the host’s overall immunity but did not reduce complications related to surgical procedures; this is consistent with previous reports [49,50,51]. Chen et al. suggested that probiotics administration may reduce postoperative infectious complications. Specifically, they found that probiotics could effectively reduce complications such as septicemia, incision infection, central line infection, pneumonia, UTI, and diarrhea [49]. However, they did not analyze procedure-related complications such as anastomotic leakage and intra-abdominal abscess formation. In addition, for one RCT included in their analysis, the incidence of septicemia was 55% and 73% in the probiotics and control groups, respectively [33]. This result is quite different from the results of other RCTs, suggesting that the definition of septicemia may be different from other studies. Ouyang et al. also reported that the application of probiotics contributed to the reduction of overall infection rate, incisional infection, and pneumonia in their meta-analysis [50]. On the other hand, similar to our results, other studies have demonstrated that probiotic administration did not influence procedure-related complications, including anastomosis leakage [50,51]. There have been several studies that have reported that there is a relationship between gut microbiota and anastomosis site healing [52,53]. However, since anastomotic leakage may be more related to the quality of surgical technique, such as the tension of the anastomotic site or perfusion of the proximal and distal colon, probiotic administration may not reduce anastomotic leakage. As such, postoperative complications are likely caused by iatrogenic injury and technical error occurring during surgery. Therefore, caution should be exercised in the interpretation of the effects of probiotics on postoperative complications.

Probiotics may reduce infectious complications through the following possible mechanisms of action. First, probiotics reduce the intestinal luminal PH, which impedes the growth of pathogenic bacteria; furthermore, it secretes antimicrobial peptides, such as human beta-defesin 2, which have direct antibacterial activity. Second, probiotics can increase mucus secretion, which prevents the adherence of pathogenic bacteria to the mucous membrane, prevents bacterial translocation, and enhances intestinal barrier function. Third, probiotics enhance immune function by increasing the activity of natural killer cells, as well as promoting the maturation of antigen-presenting cells and dendritic cells. Furthermore, probiotics promote the production of anti-inflammatory cytokines and decrease the production of pro-inflammatory cytokines, including IL-1β, IL-6, IL-8, IL-17, IL-12, tumor necrosis factor-α (TNF-α), and interferon-γ (INF- γ) [54,55].

In addition, our study showed that probiotics reduced overall postoperative complications, including non-infectious complications. Among the non-infectious complications, probiotics could effectively reduce diarrhea symptoms (RR: 0.51; 95% CI: 0.35 to 0.74), but not postoperative ileus (RR: 0.63; 95% CI 0.39 to 1.02). Other studies have shown that probiotics can effectively reduce diarrhea; this is based on the theoretical background that probiotics are effective in alleviating diarrhea by normalizing the unbalanced microflora induced by preoperative bowel cleansing and intraoperative direct intestinal manipulation [49,56,57,58,59].

In our meta-analysis, the probiotics group showed a reduced hospital LOS, with a mean difference of 1 day; however, this difference may be not clinically important for the patients based on the MCID (2 days), which is consistent with previous reports [15,28,31,34,37,56,58].

In our study, three RCTs assessed postoperative QOL by using different questionnaires [24,27,32]. Pellino et al. [32] used the SF-36 questionnaire to evaluate the QOL every week for 4 weeks after probiotics administration. They found that the probiotics group showed significantly higher scores than the control group in only one category regarding social functioning. Park et al. [24] used the EORTC QLQ-C30 to evaluate postoperative QOL and mentioned no significant between-group difference in the QOL. However, these two RCTs could not be included in the meta-analysis because statistical data available for meta-analysis, including the exact value of questionnaires, standard deviation, and *p*-value, were missing. Therefore, we only analyzed the results reported by Theodoropoulos et al. [27], who found that probiotics may have no or little effect on postoperative QOL considering MCID (6.5 points) [48]. Taken together, since few studies have investigated the effects of probiotics on the QOL after colorectal cancer surgery and given the among-study differences in the QOL questionnaires, we believe that ours is the first systematic review reporting the QOL outcome.

Studies on the effects of probiotics have been actively conducted not only for colorectal cancer but also for other diseases. An RCT on patients with gastric adenocarcinoma who underwent radical gastrectomy after preoperative chemotherapy showed that the probiotics group had significantly lower overall infectious complications, hospital LOS, and time to first flatus than the placebo group [60]. Moreover, an RCT of patients who underwent pancreatic surgery for periampullary neoplasm showed that the synbiotics group had significantly lower infectious and overall complications, as well as shorter hospital LOS than the placebo group [61]. In addition, Chowdhury et al. conducted a meta-analysis on the effects of probiotics administration in patients undergoing various kinds of abdominal surgery. They concluded that probiotics reduced postoperative infectious complications even though there was heterogeneity involving several diseases (RR: 0.56; 95% CI: 0.46 to 0.69; I^2^ = 42%) [62].

### Advantage and Disadvantage

This meta-analysis had several limitations. First, there was heterogeneity in the type of probiotics and duration of probiotics administration across the studies. Second, the assessment of infectious complications may vary across studies since it requires subjective judgment. Third, we did not evaluate publication bias since most of the outcomes were reported by <10 studies; however, publication bias may exist. Fourth, there was no large-scale study meeting the optimal information size. Nonetheless, this study has strengths. Previous meta-analyses showed shortcomings in study selection, including reduced consistency of participants and interventions. Contrastingly, we performed a more thorough screening of participants, interventions, and comparators to evaluate the effect of probiotics on postoperative outcomes in patients who underwent colorectal cancer surgery. Furthermore, we analyzed with rigorous Cochrane methodologies and applied the GRADE approach to evaluate CoE.

## 5. Conclusions

Perioperative probiotic administration may have effects on reducing postoperative complications, including overall infectious complications, in patients undergoing colorectal cancer surgery without any significant adverse effects. Compared to standard of care or placebo, probiotics may have similar effects on perioperative mortality and procedure-related complications such as anastomotic leakage, hospital LOS, and QOL. Thus, probiotics may be considered a beneficial supplement to routine perioperative care for colorectal cancer surgery. However, the results of our meta-analysis were mostly based on the low CoE, and large-scale RCTs are warranted to elucidate the effect of probiotics. Finally, given the diversity in the use and types of probiotics, additional research is warranted to establish an optimal treatment protocol.

## Figures and Tables

**Figure 1 medicina-58-01644-f001:**
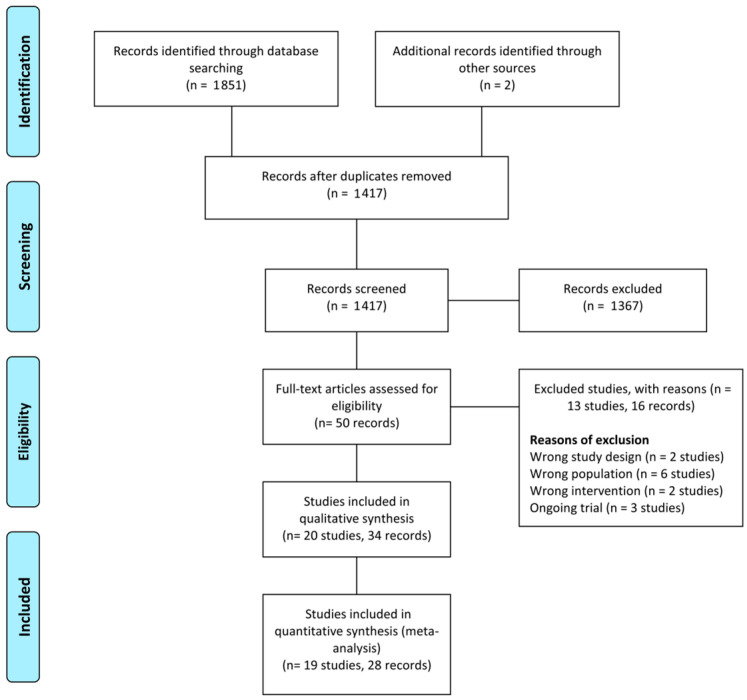
PRISMA flow diagram.

**Figure 2 medicina-58-01644-f002:**
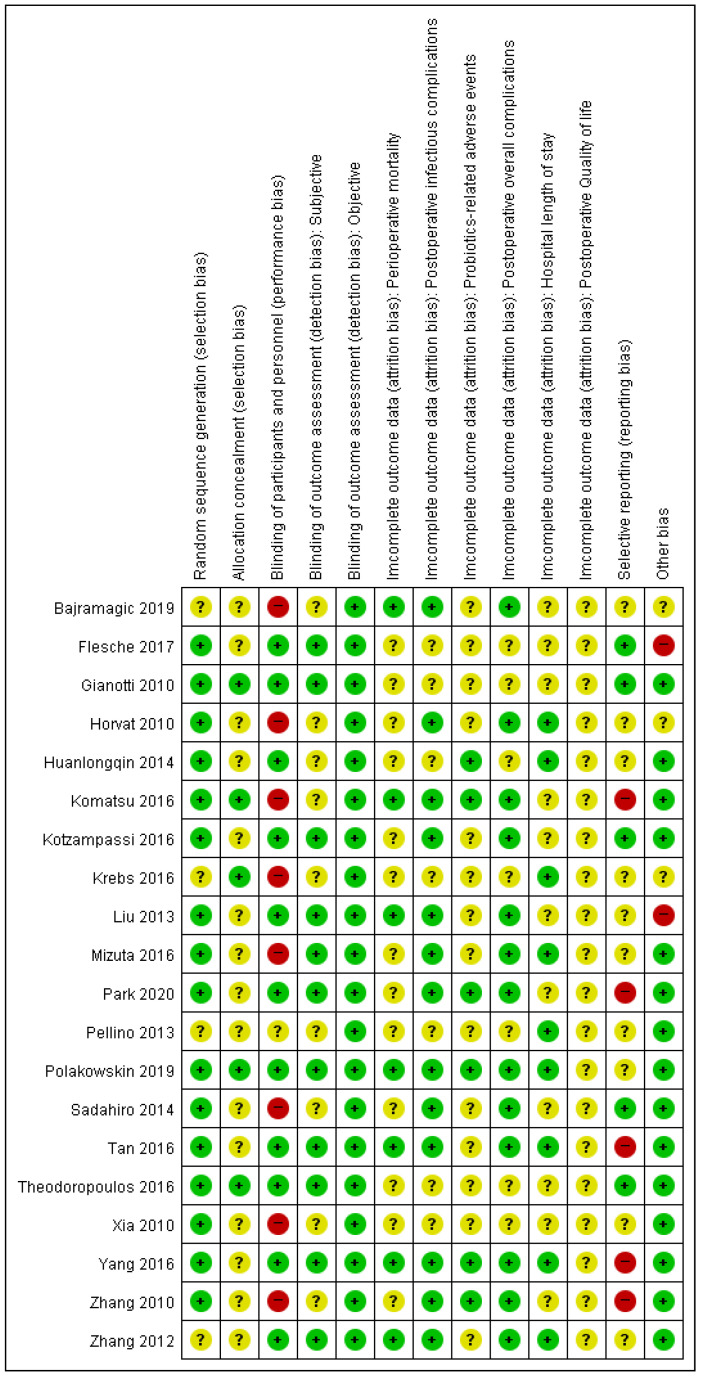
Risk of bias summary: Review authors’ judgements about each risk of bias item for each included study. Subjective outcomes: perioperative mortality and hospital length of stay. Objective outcomes: postoperative infectious complications, probiotics-related adverse events, postoperative overall complications, and postoperative quality of life. Categories: Green point (+) = low risk of bias; yellow point (?) = unclear risk of bias; red point (−) = high risk of bias.

**Table 1 medicina-58-01644-t001:** Baseline characteristics of included studies.

Study	Study Design/Setting	Trial Period (Year to Year)	Country /Language	Type of Surgery	Stage	Total Number of Analyzed Participant		Age (Mean ± Standard Deviation)		Treatment		**Route**	**Duration of Administration**	**Duration of Follow-Up (Months)**
						Intervention	Control	Intervention	Control	Intervention	**Control**			
Park 2020 [24]	RCT/multicenter	2016 to 2018	Korea/ English	sigmoid colon cancer resection (anterior resection)	I~III	29	31	60.1 ± 10.37	61.03 ± 7.02	probiotics (Bifidobacterium animalis, lactis, Lactobacillus casei, and Lactobacillus plantarum)	placebo	oral	1 week before surgery to 21 days after surgery	1 month
Polakowski 2019 [14]	RCT/single center		Brazil/ English	colorectal cancer resection	I~III	36	37	60.9 ± 6.7	58.9 ± 6.3	synbiotics (Lactobacillus acidophilus NCFM, L. rhamnosus HN001, L. casei LPC-37, and Bifidobacterium lactis HN019 + fructooligosaccharide)	placebo	oral	8 days before surgery to the day before surgery	1 month
Bajramagic 2019 [25]	RCT/single center	2017 to 2017	Bosnia/ English	colorectal cancer resection	III	39	39			probiotics (Lactobacillus acidophilus, Lactobacillus casei, Lactobacillus plantarum, Lactobacillus rhamnosus, Bifidobacterium lactis, Bifidobacterium bifidum, Bifidobacterium breve, and Streptococcus thermophilus)	standard care	oral	3 days after surgery to 30 days after surgery	12 months
Flesche 2017 [26]	RCT/single center	2013 to 2015	Brazil/ English	colorectal cancer resection	I~IV	49	42	64.5 ^a^	61.6 ^a^	synbiotics (Lactobacillus acidophilus NCFM, L. rhamnosus HN001, L. paracasei LPC-37, and Bifidobacterium lactis HN019 + oligosaccharide)	placebo	oral	5 days before surgery to 14 days after surgery	1 month
Yang 2016 [15]	RCT/single center	2011 to 2012	China/ English	colorectal cancer resection	I~III	30	30	63.90 ± 12.25	62.17 ± 11.06	probiotics (Bifidobacterium longum, Lactobacillus acidophilus, and Enterococcus faecalis)	placebo	oral	5 days before surgery to 7 days after surgery	1 month
Theodoropoulos 2016 [27]	RCT/single center	2008 to 2012	Greece/ English	colorectal cancer resection	0~IV	34	33	66.8 ± 2.17	69 ± 1.37	synbiotics (Pediococcus pentosaceus, Leuconostoc mesenteroides, Lactobacillus paracasei spp. paracasei, and Lactobacillus plantarum, and 2.5 g of each of the four fermentable fibers (prebiotics))	placebo	oral	15 days from 2 days after surgery	6 months
Tan 2016 [16]	RCT/single center	2012 to 2015	Malaysia/ English	colorectal cancer resection	I~III	20	20	64.3 ± 14.5	68 ± 11.9	probiotics (Lactobacillus acidophilus, Lactobacillus casei, Lactobacillus lactis, Bifidobacterium bifidum, Bifidobacterium longum, and Bifidobacterium infantis)	placebo	oral	8 days before surgery to the day before surgery	1 month
Mizuta 2016 [28]	RCT/single center	2008 to 2012	Japan/ English	colorectal cancer resection		31	29	68.9 ± 10.4	71.2 ± 9.5	probiotics (Bifidobacterium longum)	standard care	oral	7–14 days before surgery to 14 days after surgery	2 weeks
Krebs 2016 [29]	RCT/single center	2009 to 2012	Slovenia/ English	colorectal cancer resection		18	16	62 (43~87) ^a^	67 (52~78) ^a^	synbiotics (Pediacoccus pentosaceus, Leuconostoc mesenteroides, Lactobacillus paracasei, and Lactobacillus plantarum)	standard care	oral	3 days before the surgery	1 month
Komatsu 2016 [17]	RCT/single center	2008 to 2013	Japan/ English	laparoscopic colorectal cancer resection	0~IV	168	194	66.7 ± 11.6	67.7 ± 10.7	synbiotics (Lactobacillus casei, Bifidobacterium breve + galactooligosaccharides)	standard care	oral	7–11 days before surgery to 2–7 days after surgery	1 month
Kotzampassi 2016 [18]	RCT/single center	2013 to 2014	Greece/ English	colorectal cancer resection		84	80	65.9 ± 11.5	66.4 ± 11.9	probiotics (Lactobacillus acidophilus, Lactobacillus plantarum, Bifidobacterium lactis, and Saccharomyces boulardii)+K14	placebo	oral	the day of surgery to 14 days after surgery	1 month
Sadahiro 2014 [30]	RCT/single center	2008 to 2011	Japan/ English	colon cancer resection	I~III	100	95	67 ± 9	66 ± 12	probiotics (Bifidobacteria)	standard care	oral	7 days before surgery to 5–10 days after surgery	1 month
Huanlongqin 2014 [31]	RCT/single center	2011 to 2011	China/ Chinese	colorectal cancer resection	I~III	30	30	59.8 ± 18.7	60.3 ± 17.2	probiotics (lactic acid bacteria)	placebo	oral	5 days before surgery to 7 days after surgery	1 month
Pellino 2013 [32]	RCT/single center	2005 to 2012	Italy/ English	laparoscopic colon cancer resection		10	8	71.5 ± 2.1	72.9 ± 1.6	probiotics (Streptococcus thermophilus, Bifidobacteria, Lactobacillus acidophilus, L. plantarum, L. paracasei, and L. delbrueckii subsp. Bulgaricus)	placebo	oral	1 day after discontinuation of antibiotics to 4 weeks	1 month
Liu 2013 [33]	RCT/multicenter	2007 to 2011	China/ English	colorectal cancer resection	I~III	75	75	62.28 ± 12.41	66.06 ± 11.02	probiotics (Lactobacillus plantarum, Lactobacillus acidophilus, and Bifidobacterium longum)	placebo	oral	6 days before surgery to 10 days after surgery	1 month
Zhang 2012 [34]	RCT/single center	2006 to 2007	China/ English	colorectal cancer resection	I~III	30	30	67.5 (45.0~87.0) ^a^	61.5 (46.0~82.0) ^a^	probiotics (B. longum, L. acidophilus and Enterococcus faecalis)	placebo	oral	5 days before surgery to 3 days before surgery	1 month
Horvat 2010 [37]	RCT/single center		Slovenia/ English	colon cancer resection		20	20	62 (42~86) ^a^	65 (52~78) ^a^	synbiotics (Pediococcus pentosaceus, Leuconostoc mesenteroides, Lactobacillus paracasei subsp. Paracasei, and Lactobacillus plantarum)	standard care	oral	3 days before the surgery	1 month
Xia Yang 2010 [36]	RCT/single center	2008 to 2008	China/ Chinese	colorectal cancer resection		30	30			probiotics (sour milk, lactic acid bacteria)	standard care	oral	more than 5 days until the day before surgery	1 month
Zhang 2010 [35]	RCT/single center	2006 to 2007	China/ Chinese	colorectal cancer resection		30	30	66.7 (41~83) ^a^	63.0 (39~81) ^a^	probiotics (Bifidobacterium)	standard care	oral	for 5 days before the surgery	1 month
Gianotti 2010 [38]	RCT/muti center (two)	2006 to 2007	Italy/ English	colorectal cancer resection		21	10	63.3 ± 102	62.7 ± 7.8	Probiotics (Lactobacillus johnsonii, Bifidobacterium longum)	placebo	oral	3 days before the surgery	1 month

RCT, randomized controlled trial. ^a^ Median (range).

## Data Availability

Not applicable.

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
