# Peer review of "Perioperative Probiotics Application for Preventing Postoperative Complications in Patients with Colorectal Cancer: A Systematic Review and Meta-Analysis"

_medicina, 2022, doi:10.3390/medicina58111644_

Round 1

Reviewer 1 Report

Sanghyun An et al conducted a high-quality systematic review and meta-analysis that aimed to evaluate the effect of probiotics on postoperative outcomes in patients undergoing surgery for colorectal cancer. This review meets the PRISMA guidelines in full detail. I congratulate the authors for this work. I would like to make a few recommendations:

-The sub-headings should be removed from the abstract;

-A recommendation would be that in the discussion, the authors should also bring up the fact that many perioperative complications are iatrogenic (e.g., https://pubmed.ncbi.nlm.nih.gov/27273964/ ; https://doi.org/10.1016/j.ejogrb.2018.03.039;  DOI: 10.1016/j.jvs.2003.11.040) and many postoperative adverse events are secondary to these iatrogenic injuries.

Author Response

1. The sub-headings should be removed from the abstract

Answer: Thanks for your proper point. We removed the sub-headings from abstract as you said.

2. A recommendation would be that in the discussion, the authors should also bring up the fact that many perioperative complications are iatrogenic (e.g., https://pubmed.ncbi.nlm.nih.gov/27273964/; https://doi.org/10.1016/j.ejogrb.2018.03.039;  DOI: 10.1016/j.jvs.2003.11.040) and many postoperative adverse events are secondary to these iatrogenic injuries.

Answer: We appreciate the reviewer’s insightful comment. We totally agree with your opinions, so we added the following sentence in the discussion part. 
“As such, postoperative complications are likely caused by iatrogenic injury and technical error occurring during surgery. Therefore, caution should be exercised in the interpretation of the effects of probiotics on postoperative complications.”

Reviewer 2 Report

Thank you for giving me the opportunity to review this meta-analysis evaluating the impact of pre-operative probiotic use. This study only included randomised trials which is a strength.

The authors assessed morbidity and mortality at 30 days. Do the authors have results at 90 days post-op, which would be more informative.

In the discussion, results on the prevalence of anastomotic fistulas and abscesses appear, although these elements do not appear in either the material and methods or the results chapter. Similarly, the severity of complications is assessed according to the Dindo-Clavien classification but no results are available. Can the authors provide additional information?

Concerning postoperative infectious complications, it would be wise to separate colonic surgery from rectal surgery, which is much more prone to postoperative complications and does not have the same natural history (neo-adjuvant treatment, protective stoma).

Author Response

1. The authors assessed morbidity and mortality at 30 days. Do the authors have results at 90 days post-op, which would be more informative.  

Answer: Thank you for your valuable comments. According to your comments, I checked whether included studies reported 90 days complications, however, only 1 out of 20 studies suggested a 90-days outcome. The study by Theodoropoulos et al. presented only the results of postoperative quality of life, and GIQLI scores were presented 1 month, 3 months, and 6 months after surgery. In addition, we planned to evaluated the surgical complication with postoperative 30 days in protocol stage to prevent selective reporting, given that most of review and guideline defined the complication within 30 days as short term complication of surgery.

2. In the discussion, results on the prevalence of anastomotic fistulas and abscesses appear, although these elements do not appear in either the material and methods or the results chapter. 

Answer: Please see the response above. In protocol stage, we decided to choose patient important outcome as a review outcome and reported those as ‘postoperative complications’ outcome because separate reporting of each surgical complication may cause reporting bias. However, we tried to help reader’s understanding about complication with adding the results in discussion. 

3. Similarly, the severity of complications is assessed according to the Dindo-Clavien classification but no results are available. Can the authors provide additional information?

Answer: We appreciate the reviewer’s insightful comment. When we planned this meta-analysis, we tried to obtain the results of postoperative complications according to the Clavien-dindo classification, but all papers except for one study (Park 2020) did not provide information on the classification of complications. We emailed the authors of each study to obtain information, but did not receive a response, so we were unable to analyze the information. Therefore, we could not analyze the data according to Clavien-dindo classification.

4. Concerning postoperative infectious complications, it would be wise to separate colonic surgery from rectal surgery, which is much more prone to postoperative complications and does not have the same natural history (neo-adjuvant treatment, protective stoma).

Answer: I totally agree with your opinions. As you mentioned, colon cancer and rectal cancer have different treatment policies and different prognosis. Therefore, as stated in the method (page 4), in protocol stage, we planned to conduct sub-group analysis by dividing colon cancer and rectal cancer. However, there were no studies that separately presented the results of colon cancer and rectal cancer, and we sent an email requesting data, but did not get any data. Of the 20 studies included in our meta-analysis, 4 studies (Park 2020, Sadahiro 2014, Pellino 2013, Horvat 2010) were conducted with colon cancer patients only. However, these four studies were not meaningfully analyzed due to the small number of patients and events. The rest of the studies were conducted with colorectal cancer patients, and as mentioned earlier, there was no study that presented the results of colon cancer group and rectal cancer group separately. If more studies are conducted in the future, we think it will be meaningful to separately analyze the effects of probiotics on the postoperative results of colon cancer and rectal cancer patients.

Round 2

Reviewer 2 Report

The authors have responded point by point to the remarks and comments.